# Prevalence and Genomic Diversity of *Salmonella enterica* Recovered from River Water in a Major Agricultural Region in Northwestern Mexico

**DOI:** 10.3390/microorganisms10061214

**Published:** 2022-06-14

**Authors:** Irvin González-López, José Andrés Medrano-Félix, Nohelia Castro-del Campo, Osvaldo López-Cuevas, Jean Pierre González-Gómez, José Benigno Valdez-Torres, José Roberto Aguirre-Sánchez, Jaime Martínez-Urtaza, Bruno Gómez-Gil, Bertram G. Lee, Beatriz Quiñones, Cristóbal Chaidez

**Affiliations:** 1Centro de Investigación en Alimentación y Desarrollo A.C. (CIAD), Coordinación Regional Culiacán, Laboratorio Nacional para la Investigación en Inocuidad Alimentaria, Culiacán 80110, Sinaloa, Mexico; irvin.gonzalez@estudiantes.ciad.mx (I.G.-L.); ncastro@ciad.mx (N.C.-d.C.); osvaldo.lopez@ciad.mx (O.L.-C.); jgonzalez.219@estudiantes.ciad.mx (J.P.G.-G.); jvaldez@ciad.mx (J.B.V.-T.); jose.aguirre.dc18@estudiantes.ciad.mx (J.R.A.-S.); 2Investigadoras e Investigadores por México, Centro de Investigación en Alimentación y Desarrollo A.C. (CIAD), Coordinación Regional Culiacán, Laboratorio Nacional para la Investigación en Inocuidad Alimentaria, Culiacán 80110, Sinaloa, Mexico; jose.medrano@ciad.mx; 3Department of Genetics and Microbiology, Universitat Autờnoma de Barcelona, 08193 Bellaterra, Spain; jaime.martinez.urtaza@uab.cat; 4Centro de Investigación en Alimentación y Desarrollo A.C. (CIAD), Coordinación Regional Mazatlán, Acuicultura y Manejo Ambiental, Mazatlán 82100, Sinaloa, Mexico; bruno@ciad.mx; 5U.S. Department of Agriculture, Agricultural Research Service, Western Regional Research Center, Produce Safety and Microbiology Research Unit, Albany, CA 94710, USA; bertram.lee@usda.gov (B.G.L.); beatriz.quinones@usda.gov (B.Q.)

**Keywords:** environmental microbiology, food safety, foodborne pathogen, genomics, river water, *Salmonella*, serovars, single nucleotide polymorphisms, ultrafiltration method

## Abstract

*Salmonella enterica* is a leading cause of human gastrointestinal disease worldwide. Given that *Salmonella* is persistent in aquatic environments, this study examined the prevalence, levels and genotypic diversity of *Salmonella* isolates recovered from major rivers in an important agricultural region in northwestern Mexico. During a 13-month period, a total of 143 river water samples were collected and subjected to size-exclusion ultrafiltration, followed by enrichment, and selective media for *Salmonella* isolation and quantitation. The recovered *Salmonella* isolates were examined by next-generation sequencing for genome characterization. *Salmonella* prevalence in river water was lower in the winter months (0.65 MPN/100 mL) and significantly higher in the summer months (13.98 MPN/100 mL), and a Poisson regression model indicated a negative effect of pH and salinity and a positive effect of river water temperature (*p* = 0.00) on *Salmonella* levels. Molecular subtyping revealed Oranienburg, Anatum and Saintpaul were the most predominant *Salmonella* serovars. Single nucleotide polymorphism (SNP)-based phylogeny revealed that the detected 27 distinct serovars from river water clustered in two major clades. Multiple nonsynonymous SNPs were detected in *stiA*, *sivH,* and *ratA*, genes required for *Salmonella* fitness and survival, and these findings identified relevant markers to potentially develop improved methods for characterizing this pathogen.

## 1. Introduction

*Salmonella enterica* is recognized as a significant causative agent of human gastrointestinal infections worldwide from foodborne and waterborne sources, and non-typhoid human salmonellosis is responsible for millions of human illnesses in a typical year [1,2,3]. Among the bacterial foodborne pathogens, *S. enterica* is one of most commonly reported in Mexico, according to the National Epidemiological Surveillance System (Sistema Nacional de Vigilancia Epidemiológica de México (SINAVE)) [4]. In particular, human salmonellosis is associated with the consumption of a diverse variety of food commodities, such as various fruits, vine vegetables, leafy greens, spices, tree nuts, poultry, beef, sprouts, flour, and ready-to-eat foods [4]. The health benefits associated with eating a well-balanced diet, including fresh fruits and vegetables, have contributed to a significant increase in the consumption of fresh produce in Mexico and many other countries during the past decades [5,6,7]. However, this higher consumption of fresh and minimally processed ready-to-eat produce has been associated with an increased incidence in foodborne related outbreaks and sporadic illnesses [8]. The identification of relevant environmental reservoirs as potential sources of *Salmonella* transmission to fresh produce is thus needed as a preventive measure to ensure a safer food supply.

Although *Salmonella* commonly resides in the gastrointestinal tract of animals, this pathogen is highly persistent in aquatic environments due to its ability to prevail for long periods in soils and sediments and to successfully adapt to stressful conditions, such as fluctuating temperature and pH, desiccation, and nutrient deprivation [9,10,11]. In the subtropical environments found in the Culiacan Valley, surface waters such as rivers and streams have been reported as relevant reservoirs for *Salmonella* due to optimal growth temperatures and the availability of nutrients (glucose, N-acetyl-D-glucosamine and D-glucosamine acid), enabling the prevalence of *Salmonella* in these aquatic environments [12,13]. Additionally, several studies indicated that organic matter transported into rivers by stormwater runoff promotes an increase in *Salmonella* populations [14,15,16].

Approximately 2500 *S. enterica* serovars have been currently identified worldwide based on cell surface H- and O-antigens [17]. In Mexico, the highest incidence in human infections has been associated with *S.* Typhimurium, *S.* Enteritidis, *S.* Derby, *S.* Agona and *S.* Anatum, which are also the most predominant serovars recovered from diverse food sources [4,18]. However, these serovars, causing the majority of human illnesses, rarely coincide with those with the highest prevalence in aquatic environments, including *S.* Oranienburg, *S.* Saintpaul, *S.* Minnesota, *S.* Give, and *S.* Infantis [19,20,21] recovered from river water belonging to the Culiacan Valley in northeastern Mexico. The low environmental prevalence of clinically relevant serovars provides a challenge for accurately determining the origin of the source of contamination and the identification of conducive conditions promoting the prevalence of these serovars in aquatic environments [9,18,22]. To aid in identifying sources of contamination, the use of high-resolution sequencing methods, such as whole genome sequencing, in conjunction with bioinformatic approaches can assist with the identification of genetic markers distinguishing environmental and clinical isolates, allowing a better understanding of the genotypic profiles of *Salmonella* strains recovered from aquatic environments in Mexico [23].

To implement strategies that allow an improved detection sensitivity for *Salmonella* in agricultural environments, the present study employed an ultrafiltration system [24,25] as an effective recovery method for *Salmonella* from samples collected from the major rivers in the Culiacan Valley, an important region in Mexico for the production of fresh produce commodities for domestic and international consumption [5,6,7]. The effects of several water physicochemical parameters on *Salmonella* recovery levels were also further investigated, and genome sequencing and bioinformatics tools were subsequently employed to determine the phylogenetic relationships of the *S. enterica* serovars recovered from irrigation rivers in this important agricultural region in northwestern Mexico.

## 2. Materials and Methods

### 2.1. Collection of River Water at Locations in the Culiacan Valley, Northwestern Mexico

The Culiacan Valley in the state of Sinaloa is a subtropical region located in northwestern Mexico, and the main activity of this region is agriculture, which is supplied with water mainly by three rivers. Originating at the Sierra Madre Occidental, the Humaya River and Tamazula River are supplied with water by the Adolfo López Mateos Dam and the Sanalona Dam, respectively, and converge in the city of Culiacan to form the Culiacan River, which then flows from east to west to empty into the Gulf of California (Figure 1). These three rivers comprise the Culiacan River basin, which is located at coordinates 105°41′ and 108°4′ W and 24°24′ and 26°24′ N [26]. Eleven river sites in the Humaya River, Tamazula River, and Culiacan River throughout the Culiacan Valley were sampled during a 13-month period, between June 2018 and November 2019. The sampled sites were named sites A, B, and C for the Humaya River; sites D, E, and F for the Tamazula River; and sites G, H, I, J, and K for the Culiacan River (Table 1). The criteria employed for the selection of the sampling points was based on the following: (i) sites in agricultural areas where rural populations were present; (ii) sites where livestock and agriculture were practiced not more than 1 km away from the sampling collection site; and (iii) sites where there was some type of pluvial discharge into the river.

A total of 10 L of river water was collected in sterile polypropylene containers per sampling site each month. Each sample was collected at a depth of 20 cm and at a distance of 50 cm from the edge of the river [27], and all samples were subjected to a collection schedule, which was performed during the last days of each month. After collection, samples were immediately transported to the Microbiology Laboratories of CIAD’s National Laboratory for Food Safety [Laboratorio Nacional para la Investigación en Inocuidad Alimentaria at the Centro de Investigación en Alimentación y Desarrollo (CIAD), Culiacan, Mexico)] in refrigerated coolers at a temperature of 4–8 °C. The microbiological analysis of each sample was performed routinely within 24 h after collection.

### 2.2. Evaluation of Environmental and Water Parameters

Several physicochemical parameters were measured at each sampling site in the Culiacan Valley. In particular, water temperature (°C), pH, electrical conductivity, total dissolved solids, and salinity were recorded for each sampling site during each collection of river water with a TRACER PockeTester™ monitoring device (Model1766; LaMotte, Chestertown, MD, USA). Additionally, rainfall (mm/m^2^) and relative humidity percentage data from the closest weather station for each sampling point were also recorded using the information provided by the website https://www.tutiempo.net/ (accessed on 1 May 2018), which reports meteorological data from different stations. For the analysis of environmental data, the averages of the measurements during each month of sampling were collected.

### 2.3. Ultrafiltration Process for River Water Samples

To increase the recovery of *Salmonella* from river water, an ultrafiltration system, based on a size-exclusion method, was employed to efficiently concentrate water volumes from liters to milliliters, resulting in a final suspension of the targeted pathogen in the retentive volume [27]. Briefly, pre-sterilized Masterflex L/S^®^ high-performance precision pump tubing (Cole-Parmer, Vernon Hills, IL, USA) and a Masterflex L/S^®^ Variable-Speed pump system (Model 7528-10; Cole-Parmer) were used for all ultrafiltration experiments with settings to generate a flow rate of 2900 mL/min and to maintain pressure at 5 to 10 psi. Single-use polysulfone dialysis filters (Model Optiflux^®^ F180NR; Fresenius Medical Care, Waltham, MA, USA) were used for each sample after blocking non-specific binding by recirculating 1 L of sterile 0.1% (*w/v*) sodium hexametaphosphate (NaPP) solution (Sigma-Aldrich, St Louis, MO, USA) for 10 min. After this incubation time, the permeate port was opened to remove the blocking solution and initiate the sample filtration process. Filtration was carried out until a final volume of 400 mL was obtained. The concentrated sample was removed from the ultrafiltration system, and 500 mL of a backflushing solution (0.01% (*w/v*) NaPP, 0.01% (*w/v*) Tween 80 and 0.001% (*w/v*) Antifoam Y-30 emulsion) (Sigma-Aldrich) was added to elute microorganisms until 100 mL of solution was obtained. Finally, the concentrated sample (400 mL) and the elution solution (100 mL) were mixed to obtain 500 mL of final retentate that was used for the recovery of *Salmonella* isolates.

### 2.4. Recovery and Characterization of S. enterica Isolates by Culturing, Biochemical and Molecular Methods

A standardized analytical protocol was employed for the identification and confirmation of non-typhoidal *Salmonella* isolates using selective media, followed by biochemical and molecular confirmation [28]. For the recovery of presumptive *Salmonella* isolates, various amounts of each collected surface water sample (undiluted, 1 mL, 10 mL, 20 mL) were subjected to an enrichment step in tryptic soy broth (TSB; Becton Dickinson Bioxon, Mexico City, Mexico) by incubating at 37 °C for 24 h. After incubation, the TSB cultures were used for inoculating modified semi-solid Rappaport Vassiliadis (MSRV) agar plates with a total of six discrete 30 μL drops from each TSB enrichment, followed by incubation at 42 °C for 18 h. After incubation on MSRV agar, presumptive *Salmonella* with a white halo around the colony were selected with a sterile platinum loop and re-streaked on xylose lysine desoxycholate (XLD) agar for further incubation at 37 °C for 24 h. Isolates with a characteristic *Salmonella* colony morphology (consisting of black colonies with a whitish halo) were selected for subsequent confirmatory biochemical and molecular tests. Biochemical tests on the presumptive *Salmonella* colonies were subsequently performed using triple sugar iron agar, lysine iron agar, and urea broth and further incubated under aerobic conditions at 37 °C for 24 h, following standardized procedures [28]. The results from the multiple-tube biochemical assays were then employed to obtain an estimation of the bacterial concentration based on the most probable number (MPN) technique and expressed as MPN per 100 mL of sample, according to the protocol for quantitative sample analyses in the EPA Method 1200 [28].

Presumptive *Salmonella* colonies were subjected to a confirmatory molecular test consisting of polymerase chain reaction (PCR) amplification of the *invA* gene, which encodes an invasion protein in *Salmonella* spp. [29], by using the primers Sal1598 Forward (5′-AACGTGTTTCCGTGCGTAAT-3′) and Sal1859 Reverse (5′-TCCATCAAATTAGCGGAGGC-3′) [30]. All oligonucleotides were purchased from Sigma-Aldrich. For preparation of crude lysates for each isolate, TSB cultures were inoculated with *Salmonella* presumptive colonies grown on XLD agar and incubated for 24 h at 37 °C under aerobic conditions. One milliliter of the overnight TSB culture was subjected to centrifugation at 11,300× *g* for 5 min, and the cell pellet was washed twice and resuspended in 200 µL of molecular biology-grade water (UltraPure, ThermoFisher, Waltham, MA, USA) for further incubation at 98 °C for 10 min in a ThermoMixer (Eppendorf Latin America, Mexico City, Mexico), followed by 10 min incubation on ice. The cell lysates were subjected to subsequent centrifugation at 6700× *g* for 5 min, and the supernatants (100–150 µL) were stored at −20 °C until further use as templates for PCR. PCR amplification reactions consisted of 2×GoTaq^®^ Green Master Mix (Promega Corporation, Madison, WI, USA), Sal1598 and Sal1859 primers (1 µM each), and 2 μL of bacterial lysate in a final volume of 25 μL for incubation in a Mastercycler^®^ thermal cycler (Eppendorf Latin America) with the following parameters: 1 cycle at 95 °C for 5 min, followed by 26 cycles of 1 min at 95 °C, 1 min at 63 °C, 1 min at 72 °C, and a final extension cycle at 72 °C for 5 min [30]. The PCR amplicons were separated on a 1% agarose gel and visualized by staining with 1 μL of 10 mg/mL Gel Red^®^ Nucleic Acid Stain (Sigma-Aldrich). Positive samples were selected based on the amplification of a 244 bp fragment of the *invA* gene [30]. All PCR-positive *Salmonella* isolates were selected and subsequently preserved at −80 °C in a 50/50 mixture of glycerol and TSB.

### 2.5. Genome Sequencing and Assembly of the Recovered S. enterica Isolates

For whole genome sequencing, the recovered *Salmonella* isolates were grown for 24 h in TSB at 37 °C under aerobic conditions, and the genomic DNA was extracted using a DNeasy Blood and Tissue kit (QIAGEN, Mexico City, Mexico) following the manufacturer’s instruction, that was initially assessed and quantified with a NanoDrop 2000c Spectrophotometer (Thermo Fisher Scientific, Waltham, MA, USA). For the sequencing reactions, genomic DNA from each isolate was then quantified with a Qubit™ 2.0 Fluorometer (Invitrogen, Carlsbad, CA, USA), adjusted to 0.2 ng/μL, and 1 ng aliquot was used for preparing genomic DNA libraries with the Nextera XT DNA Library Preparation Kit (Illumina Inc., San Diego, CA, USA). Subsequently, the prepared genomic libraries from the *Salmonella* isolates were sequenced using a MiSeq™ Reagent Kit v2 (300-cycle format) to obtain 2 × 150 bp paired-end reads with a MiSeq™ System (Illumina, Inc.) at the Earlham Institute (Norwich Research Park, Norwich, UK) or a MiniSeq™ System (Illumina, Inc.) at the Molecular Diagnostics Laboratory (Laboratorio de Diagnóstico Molecular, Centro de Investigación en Alimentación y Desarrollo, Mazatlán, Mexico) (see Appendix A). Prior to genome assembly, fastp 0.20.1 [31] was used to assess and filter the quality of the *Salmonella* genomic sequences and ensure the removal of the trimmed adapter sequences from the 3′-end of the reads. The genome assembly pipelines A5-miseq [32] and SPAdes v3.13.0 [33] were used with default parameters to assemble the genomes for isolates sequenced by the MiSeq™ and MiniSeq™ Systems, respectively.

### 2.6. In Silico Serotyping and Phylogenetic Relationships of the Recovered S. enterica Isolates

The *Salmonella* in silico typing resource (SISTR) v1.0.2 [34] was employed to perform serovar predictions of the assembled genome sequences for each *Salmonella* isolate. The default parameters were specified for using a combination of serogroup-specific probes and core genome multilocus sequence typing (cgMLST) analysis for serovar prediction. Analysis of the diversity in the detected serovars was assessed by calculating Simpson’s diversity index, where high diversity in the population is indicated by diversity index values close to 1 [35,36]. Furthermore, the relationships of the isolates were investigated through the identification of single nucleotide polymorphisms (SNPs) in the core genome by employing Parsnp v1.2, available in the Harvest software suite [37], where *Salmonella enterica* subsp. *enterica* serovar Typhimurium strain LT2 (GenBank Accession No. AE006468.2) was used as the reference strain. A maximum-likelihood phylogenetic tree was constructed using the Randomized Axelerated Maximum Likelihood (RAxML) v8 program [38] with the General-Time-Reversible (GTR) model plus gamma distribution as the nucleotide substitution rate model [39,40] using a bootstrap number of 100. The constructed phylogenetic tree was then visualized and annotated using the Interactive Tree Of Life (iTOL) v6 web-based tool [41]. For conducting the subsequent analysis based on single nucleotide polymorphisms (SNP) phylogeny and multilocus sequence typing (MLST), the assembled genomes were submitted to the pipeline CSIPhylogeny version 1.4 [42] and the pipeline multilocus sequence typing (MLST) version 2.0.4 to obtain sequence types (ST) [43] (both available on the website for the Center for Genomic Epidemiology http://www.genomicepidemiology.org/ (accessed on 1 March 2022) by using default parameters and *S.* Typhimurium strain LT2, listed above, as the reference strain. To further characterize the variable and high-quality single nucleotide polymorphisms (SNPs) identified in the recovered *S. enterica* sequenced genomes, an alignment file consisting of 131,259 SNPs was generated and subsequently visualized with GrapeTree [44] software to generate a minimum spanning tree. The nodes on the minimum spanning tree were numbered by using GrapeTree software and the list of the variable SNPs contributing to the branches are listed in the Appendix A. Branch lengths of less than 400 SNPs in the minimum spanning tree were collapsed to create a single node per serovar identified in the recovered *S. enterica* isolates.

### 2.7. Statistical Analysis

All statistical analyses were performed with Minitab^®^ 18 statistical software (Minitab, LLC, State College, PA, USA) to determine the Pearson linear correlation coefficients [14] and Poisson regression analyses [45] for describing the relationships between water and environmental parameters and the prevalence of recovered *Salmonella* levels during each sampling month. For all measures of association, *p* values <0.05 were considered statistically significant.

## 3. Results

### 3.1. Salmonella Recovery from River Sampling Locations in the Culiacan Valley, Northwestern Mexico

The present study employed a size-exclusion ultrafiltration method [24] followed by an enrichment step with selective media [28] to enable improved detection of *Salmonella* when present at low concentrations in water samples. Presumptive *Salmonella* spp. isolates were recovered from a total of 54.5% (78/143) of the river water samples collected from 11 river sampling sites in the Culiacan Valley from June 2018 to November 2019 (Figure 1 and Table 1). The highest percentage of positive water samples was obtained from sites B and D with 69.2% (9/13), followed by sites A, C, E, G, and I with 61.5% (8/13) of positive samples. Sites J and K had the lowest prevalence with 30.8% (4/13) and 23.1% (3/13) positive samples, respectively. Based on the number of combined positive samples per river, the highest number of positive samples was identified for sites along the Humaya River with 64.1% (25/39), followed by the Tamazula River with 61.5% (24/39), and then by the Culiacan River with 44.6% (29/65) of the samples.

As shown in Figure 2, quantitation of *Salmonella* in the collected river water samples was determined by employing the most probable number technique [28]. A spatial examination of the recovered *Salmonella* levels revealed higher than average recovery levels for sites B and H on the Humaya and Culiacan rivers, followed by sites A, D, and F on the Humaya and Tamazula rivers (Figure 2a). The lowest detected levels of *Salmonella* were observed for Site K on the Culiacan River prior to discharging into the Pacific Ocean (Sea of Cortez). Furthermore, temporal analysis of the recovered *Salmonella* showed that the detected levels were highly variable but significantly higher during the summer months when compared to other months of sampling (*p* = 0.000) (Figure 2b). Throughout the entire sampling period, the detectable levels of *Salmonella* in the river water samples had mean values ranging from <0.64 MPN/100 mL to 13.98 MPN/100 mL (Figure 2b).

In particular, the highest levels of *Salmonella* were detected during the summer months of June, July and August of 2018 with average value of 13.8 MPN/100 mL, 7.88 MPN/100 mL and 6.75 MPN/100 mL, respectively (Figure 2b). The lowest levels were observed from December 2018 to March 2019 and September 2019 to November 2019, corresponding to the fall and winter seasons in the Culiacan Valley with levels ranging from 0.65 MPN/100 mL to 0.93 MPN/100 mL (Figure 2b). During the summer of 2019, the *Salmonella* levels were 2.41 and 2.33 MPN/100 mL in July and August, respectively. These recovery levels were three-fold lower when compared to those detected in the summer months of the previous year but at least two-fold higher than the levels detected during the winter and fall of 2019.

### 3.2. Effect of Environmental and Physicochemical Water Parameters on Salmonella Detection Levels

The effects of several environmental parameters on the recovery levels of *Salmonella* from the river water were further evaluated. During the sampling period in the present study, the seasonal values for rainfall ranged from 3.0 to 194.6 mm/m^2^, and the recorded relative humidity fluctuated from 51.09 to 75.5% (Appendix A). The lowest values for these environmental parameters were recorded in the fall and winter months (October to April), whereas the highest values were registered in summer (June to September). It is noteworthy that rainfall levels exceeded 214 mm/m^2^ in September 2018 due to Tropical Depression 19-E, which caused flooding throughout the Culiacan Valley [46]. Further analysis of physicochemical water parameters revealed that the river water had temperatures ranging from 21.8 °C to 36.4 °C (Appendix A). Additional examination of other physicochemical water parameters for sites A to I revealed similar values throughout the year for electrical conductivity (<613.46 µS/cm), total dissolved solids (<387 ppm), and salinity (<0.29 PSU). As expected, the farthest site, K, located on the Culiacan River and in close proximity to the Pacific Ocean, had the highest mean values for electrical conductivity (6477 µS/cm), total dissolved solids (6066 ppm), and salinity (9 PSU) (Appendix A). When compared to all other sampling sites, site J on the Culiacan River showed the greatest fluctuations in the recorded levels for electrical conductivity (4130–6400 µS/cm), total dissolved solids (2510–4950 ppm), and salinity (1.9–2.76 PSU) during the months of January to July. According to Pearson’s analysis, the physicochemical parameters, such as total dissolved solids, electrical conductivity, and salinity showed a high positive correlation, indicating a statistically significant (*p* < 0.05) linear relationship with each other (Table 2). Moreover, rainfall and relative humidity also had a significant positive correlation (*r* = 0.762, *p* = 0.000). Further analysis of the correlations for the various environmental parameters with the detected *Salmonella* levels revealed that only the river water temperature showed a significant positive correlation (*r* = 0.204, *p* = 0.017) (Table 2): therefore, at higher or lower temperatures, *Salmonella* levels increased or decreased, respectively. By contrast, the river water pH was found to be negatively correlated with the levels of *Salmonella* in the Culiacan Valley (*r* = −0.231, *p* = 0.007).

To further quantify any significant correlations between the physicochemical parameters and the enumerated *Salmonella* spp. from river water, a Poisson regression model was performed (Table 3). The results indicated that only the environmental data recorded during June, July, and August were significantly associated with an increase in *Salmonella* levels (*p* = 0.000). In particular, the month of June in the first year of sampling showed the highest levels of *Salmonella*, and when compared to the other months of sampling, the levels of *Salmonella* in June were observed to have a significant log increase of e^2.845^ = 17.2 (95% confidence interval (CI): 12.94–22.87) (Table 3). In the month of July for both sampling years, a log increase in *Salmonella* concentrations was calculated as e^1.371^ = 3.94 (95% CI: 2.8–5.53), followed by a gradual decrease in the month of August with an average level of e^1.303^ = 3.68 (95% CI: 2.61–5.21). Furthermore, the results indicated that for each unit increase of pH and salinity, the average of concentration of *Salmonella* spp. decreased by e^−0.397^ = 0.67 (95% CI: 0.487–0.923) and e^−0.115^ = 0.89 (95% CI: 0.835–0.951), respectively (Table 3). Based on these quantitative analyses, only two physicochemical water parameters, pH and salinity, were found to have a significant quantitative negative effect on the detected levels of *Salmonella* recovered from river water in the Culiacan Valley.

### 3.3. Serovar Identification and Phylogenetic Relationships among the Recovered Salmonella Isolates from River Water in the Culiacan Valley

To perform a genotypic classification of the *Salmonella* isolates recovered from the river sampling sites in the Culiacan Valley, a total of 73 *Salmonella* isolates were subjected to whole genome sequencing (Appendix A). The sequencing results revealed that 100% (73/73) of the recovered isolates were classified as *S. enterica* subspecies *enterica*. Subsequent in silico identification of serovar-specific genes in *S. enterica* subspecies *enterica* revealed 27 distinct serovars, and the isolates belonging to serovar Oranienburg were the most prevalent, representing 23% (17/73) of the recovered isolates. Moreover, lower prevalences were detected for *S.* Anatum (8% (6/73)), *S.* Saintpaul (7% (5/73)), *S.* Pomona (5% (4/73)), and *S.* Sandiego (5% (4/73)). *S.* Oranienburg had the highest distribution and was detected in 90.9% (10/11) of the sampling sites, followed by *S.* Anatum, and *S.* Sandiego, which were detected in 4 of the 11 sampling sites (Appendix A). The highest diversity in serovar distribution was detected in samples collected from the Tamazula and Culiacan Rivers, with a total of 16 serovars identified. The lowest diversity was detected in the *S. enterica* isolates recovered from the Humaya River, with 11 distinct serovars. The diversity in the recovered serovars was examined by calculating Simpson’s diversity index, and the results from this quantification revealed high diversity index values of 0.92 for the Humaya River, 0.95 for the Tamazula River, and 0.90 for the Culiacan River. Further quantification of Simpson’s diversity index for the serovars recovered from all three rivers revealed a diversity index of 0.91 during the summer months of June, July and August as well as a high diversity index of 0.95 for the remaining non-summer months.

To further analyze the genetic relatedness of the isolates, the *S. enterica* genomes were subjected to MLST as well as a phylogenetic analysis based on the identification of each isolate’s core genome of SNPs as a method for achieving high resolution genotyping and for classifying isolates based on different polymorphisms (Figure 3).

The typing analysis by MLST demonstrated that the ST of each isolate corresponded to those belonging to the same serovar except for *S.* Minnesota, where one isolate had an ST of 285 and the other isolate had an ST of 548 (Appendix A). Moreover, the SNP analysis revealed that the recovered *S. enterica* isolates from river water could be classified in two major clades (Figure 3). Most of the recovered *S. enterica* isolates clustered in clade 1, significantly represented by *S.* Oranienburg, the predominant serovar identified in the present study. Clade 2 consisted of only 38% of all isolates, mostly recovered from the Humaya River, where Anatum and Saintpaul were the most common serovars (Figure 3). Sequence analysis of the core genome revealed *S. enterica* isolates belonging to the same serotype and recovered from different rivers were closely related and only differed by 400 SNPs. By contrast, isolates belonging to different serovars within the same clade were found to vary by approximately 15,000 to 20,000 SNPs. One notable exception concerns the *S.* Bovismorbificans isolates, belonging to clade 1, which were found to differ by approximately 30,000 SNPs from those isolates belonging to both clades.

As shown in Figure 4, a subsequent validation of all SNPs detected in the core genome for all sequenced *S. enterica* isolates was conducted with a novel SNP analysis method employing the CSIPhylogeny pipeline (Kaas 2014). In agreement with the core genome SNP analysis (Figure 3), results from the SNP-based phylogeny revealed that both synonymous and nonsynonymous SNPs in the sequenced *S. enterica* isolates clustered in two separate clades (Figure 4). Serovars belonging to clade 1 were more closely related than of those in clade 2 (Figure 4a). All three sampled rivers were found to be well represented in both clades (Figure 4b).

From the analysis of 600 SNPs that distinguished clade 1 from clade 2, a total of 62 nonsynonymous SNPs, affecting protein function, were identified (Appendix A). Further analysis of the detected nonsynonymous SNPs revealed that the vast majority, corresponding to 48% (30/62) of the polymorphisms, reside in genes conferring virulence traits associated with adhesion, motility, and colonization of the host, as well as survival from nutritional and oxidative stress (Appendix A). Other SNPs were found in genes responsible for metabolic traits, motility, and membrane function (Appendix A). Interestingly, five of the detected SNPs affected *stiA*, a starvation-inducible loci associated with survival during prolonged nutritional starvation due to deficiencies of phosphate, carbon, and nitrogen [47]. Other multiple SNPs were identified in the virulence genes *ratA* and *sivH*, both located on the *Salmonella* pathogenicity island CS54 and required for intestinal colonization in the host [48].

## 4. Discussion

The present study examined the prevalence and diversity of *S. enterica* serovars recovered from major rivers in the Culiacan Valley, a major fresh produce production region in northwestern Mexico. Subsequent analysis evaluated physicochemical water parameters and their influence on the prevalence of *Salmonella* in rivers in the Culiacan Valley. This fundamental information will aid the identification of potential contamination sources for the development of focused and efficient intervention strategies for mitigating outbreaks caused by this foodborne pathogen. In Mexico, the National Epidemiological Surveillance System (Sistema Nacional de Vigilancia Epidemiológica, SINAVE) demonstrated a considerable increase in cases of salmonellosis from 1984 to 2017 (from 31,943 to 104,471), and the state of Sinaloa had the highest rate corresponding to 12.9% of cases [49]. Although the routes of exposure to *Salmonella* in this agricultural region have not been well defined and are often epidemiologically unrelated to specific sources [18,20,27], several studies have suggested that surface waters in this agricultural region could potentially serve as the source of *Salmonella* contamination of produce. Therefore, tracing *Salmonella* in surface waters is of utmost importance, not only for human health, but also for the safety of the fresh produce industry in this region [18,27,50].

The Adolfo López Mateos and Sanalona dams supply water to three rivers in the Culiacán Valley region, where it is used for agricultural, recreation, and domestic purposes. Some surrounding communities do not have efficient wastewater treatment, leading to wastewater mixing with surrounding waters. Additionally, agriculture, livestock, and wildlife animals, which are prevalent in this region, can be potential sources of *Salmonella*, where the main source of water for this agricultural region are provided by the Humaya, Tamazula and Culiacan Rivers [51,52]. In the present study, the effect of environmental factors on the detected levels of *Salmonella* in this agricultural region were examined by using an ultrafiltration system to increases the probability and percentage of *Salmonella* recovery, which can have a great impact on the analysis of *Salmonella* levels [27]. Recent findings have indicated that the prevalence and survival of *Salmonella* in surface waters may be due, in part, to the temperature, rainfall, as well as the physicochemical conditions of the water, and even the type of hosts that coexist in this type of habitat [53,54]. In agreement with previous studies demonstrating that salmonellosis cases reaches its peak in the summer months worldwide [27,55,56,57], the present study reported that the highest levels of *Salmonella* were observed in the summer months (June, July, and August) with concentrations significantly higher than the other months.

Among the environmental parameters examined in this study, only temperature was positively correlated with *Salmonella* levels, indicating warm temperatures may facilitate higher environmental persistence of the pathogen by promoting an increase in metabolic activity during the summer months [45,54,58,59]. By contrast, the levels of precipitation did not positively correlate with the detected levels of *Salmonella*, and the concentrations of *Salmonella* gradually decreased as a consequence of an increase in precipitation levels. Recent evidence has shown rainfall can affect *Salmonella* levels by causing the dilution of microbial populations [14,60,61,62]. During the survey in this study, precipitation levels in the Culiacan Valley reached 214.2 mm/m^2^ due to the presence of storms [46,51], and the high amounts of rainfall may have significantly affected the prevalence of *Salmonella* during the following months. Among the other physicochemical parameters for the river water, only pH and salinity showed a significantly negative association with *Salmonella* levels. The sampling sites J and K, which were in close proximity to the Pacific Ocean, had the lowest *Salmonella* levels and the highest levels of salinity. In agreement with previous studies, high levels of salinity harbor a low prevalence of *Salmonella* [22,59,63,64].

In this survey, a total of 27 *S. enterica* serovars were recovered from the rivers in the Culiacan Valley. Further quantitative analysis determined a Simpson’s diversity index with combined values higher that 0.92, revealing that a high diversity of serovars was found in this survey. The diversity identified in this agricultural region in Mexico is significantly higher than that reported for other major agricultural regions for fresh produce in the United States, which have an overall index of 0.89 [65]. Although the clinically associated serovar Typhimurium was recovered at low levels during the sampling period in the Culiacan Valley, *S.* Oranienburg, *S.* Anatum, *S.* Saintpaul, and *S.* Pomona were commonly identified. Interestingly, *S.* Oranienburg was consistently recovered from most sampling sites and predominantly detected during the summer months. These findings are in agreement with previous studies reporting that *S.* Oranienburg was one of the most prevalent serovars in aquatic environments and animal hosts in northwest Mexico [21,27,50]. Recently, *S.* Oranienburg was identified as the causative agent in a salmonellosis outbreak in Mexico associated with the consumption of fresh produce [66], highlighting the need for close monitoring of this serovar in agricultural regions. According to the SNP phylogenetic analysis conducted in the present study, *S. enterica* isolates belonging to same serovar clustered together by MLST and had similar phylogenies according to the analysis of core genome SNPs and highly variable SNPs. Recently, a report on *S. enterica* isolates recovered from surface waters in central Mexico identified significantly higher intraserovar diversity based on the CSIPhylogeny analysis of highly variable SNPs [23]. Moreover, SNP phylogeny identified that some isolates belonging to the same serovar were grouped in different SNP clusters [23]. By contrast, the *S. enterica* isolates from river water in the Culiacan Valley in northwestern Mexico were closely related with lower intraserovar diversity and were found to have less than 400 SNP if they belonged to the same serovar. Importantly, isolates belonging to different serovars were found to differ by over 10,000 SNPs. To further characterize the differences among serovars, the SNP-based phylogeny for the recovered *S. enterica* isolates identified several nonsynonymous polymorphisms in the virulence gene *stiA*, a locus that is *rpoS*-dependent and required for stress situations, including thermotolerance, acid tolerance, and survival from carbon starvation [47]. Other SNPs were identified in *sinH*, also known as *sivH*, and *ratA*, which are both found in the 25-kb genetic island at centisome 54 (CS54 island) in *S.* Typhimurium and are required for persistent colonization of the animal host [48,67]. Future work will be aimed at characterizing relevant markers contributing to the fitness and survival of *S. enterica* in aquatic environments in the Culiacan Valley, an important agricultural region in northwestern Mexico. The recovery of *S. enterica* serovars from agricultural rivers warrants a more detailed gene expression analysis that will lead into the identification of fitness traits conferring increased survival in surface water habitats.

## Figures and Tables

**Figure 1 microorganisms-10-01214-f001:**
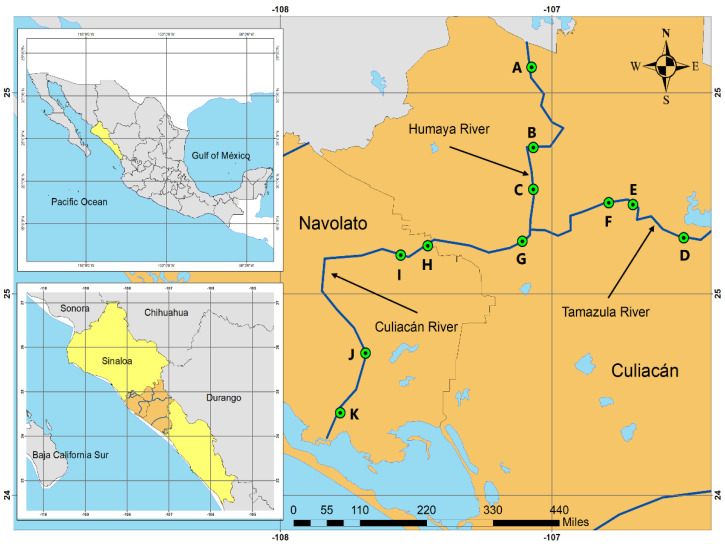
Schematic diagram of the sampling sites in the Culiacan Valley in northwestern Mexico. Sampling sites A, B, and C were located along the Humaya River, and sampling sites D, E, and F were located along the Tamazula River. Site G is located at the point of convergence for both rivers and along with sites H, I, J and K, they belong to the Culiacan River.

**Figure 2 microorganisms-10-01214-f002:**
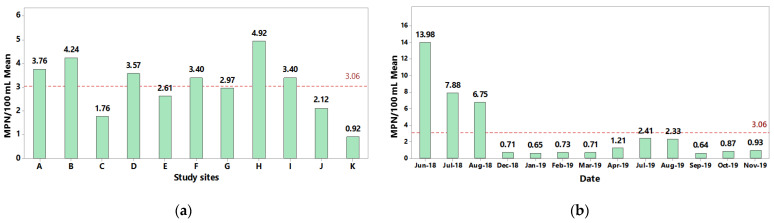
Estimation of *Salmonella* levels recovered from river water samples. A standardized analytical protocol employing culturing, biochemical, and molecular confirmation [28] was used to determine the *Salmonella* levels based on the most probable number (MPN) technique per 100 mL of sample for the various sites (**a**) and dates (**b**). The red dashed line represents the overall mean of *Salmonella* levels for all sampling sites and dates.

**Figure 3 microorganisms-10-01214-f003:**
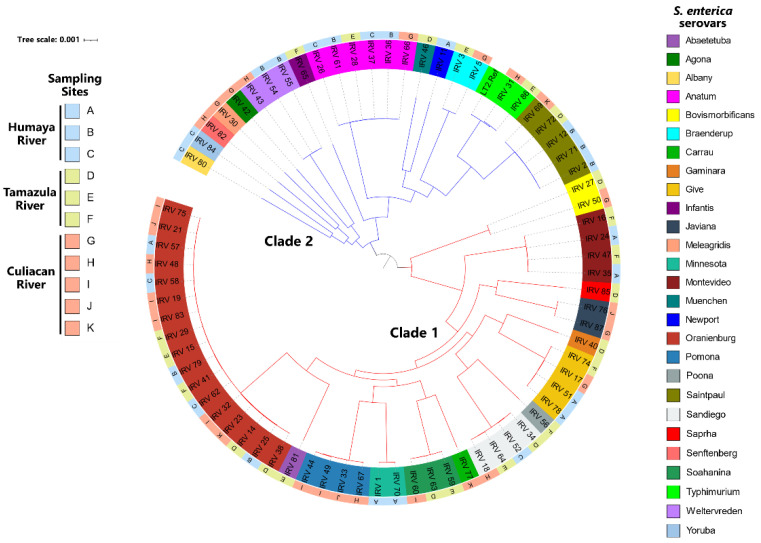
Phylogenetic tree of *S. enterica* serovars isolated from rivers in the Culiacan Valley, northwestern Mexico. A maximum-likelihood phylogenetic tree was constructed from core genome SNPs using RAxML v8 [38] under the General-Time-Reversible plus gamma distribution model with a boot strap value of 100. The constructed phylogenetic tree was visualized and annotated using the Interactive Tree of Life (iTOL) v6 web-based tool [41], and two clades are indicated by the red and blue branches.

**Figure 4 microorganisms-10-01214-f004:**
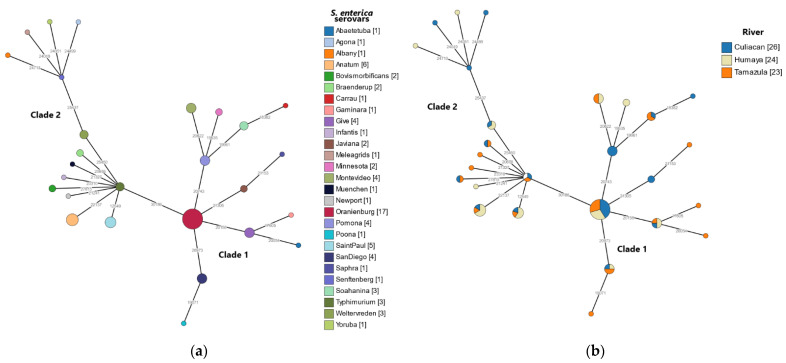
Minimum spanning tree of SNPs identified in the *S. enterica* isolates from river water samples in the Culiacan Valley, northwestern Mexico. An alignment of highly variable SNPs identified in the genomes of the recovered *S. enterica* isolates was generated using CSIPhylogeny version 1.4 [42]. The alignment was visualized using GrapeTree software with the MSTree V2 algorithim [44] to generate a minimum spanning tree. SNPs clusters were colored according to the detected serovar (**a**) or the sampled river (**b**) in the Culiacan Valley. Nodes less than 400 SNPs apart were collapsed to form a single SNP cluster, and the node size is indicative of the number of isolates.

**Table 1 microorganisms-10-01214-t001:** Latitude and longitude coordinates of each sampling site for the recovery of water from the Humaya River, Tamazula River, and Culiacan River in northwestern Mexico.

River	Sampling Site	Location Name	Sampling Site Latitude	Sampling Site Longitude
Humaya	A	Adolfo López Mateos Dam	25°02′46.9″	−107°23′50.5″
	B	Agua Caliente	24°55′44.2″	−107°23′14.9″
	C	La Guásima	25°52′10.2″	−107°24′37.1″
Tamazula	D	Sanalona Dam	25°41′54.8″	−108°38′41.3″
	E	Imala	24°51′11.7″	−107°13′17.2″
	F	Las Peñitas	24°51′41.8″	−107°15′21.1″
Culiacan	G	Puente Negro	24°48′24.3″	−107°24′34.4″
	H	San Pedro	24°47′06.2″	−107°33′31.7″
	I	Cofradía de San Pedro	24°46′47.3″	−107°35′39.8″
	J	Iraguato	24°37′39.2″	−107°39′39.2″
	K	El Castillo	24°32′39.7″	−107°42′22.8″

**Table 2 microorganisms-10-01214-t002:** Pearson’s correlation coefficients (*r*) and *p*-values (*p*) determined for the environmental parameters and *Salmonella* detection levels for all sampling sites throughout the sampling period.

Environmental Parameter	*Salmonella* Levels ^a^		pH		River Water Temperature		Total Dissolved Solids		Salinity		Electrical Conductivity		Rainfall	
	*r* ^b^	*p*	*r*	*p*	*r*	*p*	*r*	*p*	*r*	*p*	*r*	*p*	*r*	*p*
pH	−0.231	0.007												
River water temperature	0.204	0.017	0.173	0.044										
Total dissolved solids	−0.143	0.097	0.219	0.010	0.100	0.246								
Salinity	−0.133	0.123	−0.236	0.006	0.106	0.221	0.840	0.000						
Electrical conductivity	−0.143	0.097	0.215	0.012	0.095	0.269	0.985	0.000	0.861	0.000				
Rainfall	0.056	0.518	−0.012	0.890	0.767	0.000	−0.030	0.731	−0.027	0.753	−0.035	0.685		
Relative humidity	−0.049	0.572	0.069	0.426	0.561	0.000	0.045	0.605	0.048	0.582	0.036	0.681	0.762	0.000

^a^ Estimation of *Salmonella* levels based on the most probable number (MPN) technique per 100 mL of sample [28]. ^b^ Pearson’s correlation coefficients displayed in bold are statistically significant (*p*-value < 0.05).

**Table 3 microorganisms-10-01214-t003:** Parameters examined with Poisson regression analysis for *Salmonella* detection levels recovered from river water in northwestern Mexico.

Environmental Parameter	Regression Coefficient (*e*^x^)	Standard Error Coefficient	Regression Coefficient 95% Confidence Interval (CI)	*Z*-Value	*p*-Value	Variance Inflation Factor
Constant	2.954	1.264	(0.48, 5.43)	2.34	0.019	
*Salmonella* levels in June ^a^	2.845	0.147	(2.56, 3.13)	19.40	0.000	1.80
*Salmonella* levels in July	1.371	0.175	(1.03, 1.71)	7.86	0.000	1.53
*Salmonella* levels in August	1.303	0.177	(0.96, 1.65)	7.35	0.000	1.52
pH	−0.397	0.163	(−0.72, −0.08)	−2.43	0.015	1.11
Salinity	−0.115	0.035	(−0.18, −0.05)	−3.29	0.001	1.07

^a^ Estimation of *Salmonella* levels based on the most probable number (MPN) technique per 100 mL of sample [28].

## Data Availability

The complete genome sequences of the 73 *S. enterica* isolates have been deposited with links to BioProject accession number PRJNA832240 in the National Center for Biotechnology Information (NCBI) BioProject database (https://www.ncbi.nlm.nih.gov/bioproject/ (accessed on 1 May 2022). Accession numbers for each *S. enterica* isolate are found in the Appendix A.

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
