# Peer review of "Prevalence and Genomic Diversity of Salmonella enterica Recovered from River Water in a Major Agricultural Region in Northwestern Mexico"

_microorganisms, 2022, doi:10.3390/microorganisms10061214_

Round 1

Reviewer 1 Report

This is an interesting article that examines the prevalence, and the genotypic diversity of Salmonella isolates from major rivers in Northwest, Mexico. Genome sequencing and bioinformatics tools were employed to determine the phylogenetic relationships of the S. enterica serovars recovered.

The topic is current since Salmonella is one of the main causes of foodborne disease worldwide. The paper is well written, and the results should be useful for the scientific community. For all these reasons, this paper is, in my opinion, suited for publication. Moreover, there are some needed minor revisions: 

L. 90-94: Please add some references about the relationship between physicochemical parameters and genetic diversity, e.g. “Construction of Listeria monocytogenes mutants with in-frame deletions in putative ATP-binding cassette (ABC) transporters and analysis of their growth under stress conditions, Liu et al 2012.

L. 194-205 Please, add some references about methods, e. g. Norovirus detection in shellfish using two real-time RT-PCR methods, Suffredini et al 2011.

Author Response

REVIEWER 1

  1. REVIEWER’S COMMENT - L. 90-94: Please add some references about the relationship between physicochemical parameters and genetic diversity, e.g. “Construction of Listeria monocytogenes mutants with in-frame deletions in putative ATP-binding cassette (ABC) transporters and analysis of their growth under stress conditions, Liu et al 2012.

AUTHORS’ REPLY - Lines 90-94: The relevant publications by Clemens, R.L. 2015; Flores, D. 2018; Huang, S.W. 2013 were cited in lines 90-94 since the statement was describing the importance of the studied region in Mexico for the production of agricultural commodities. 

  1. REVIEWER’S COMMENT - L. 194-205 Please, add some references about methods, e. g. Norovirus detection in shellfish using two real-time RT-PCR methods, Suffredini et al 2011.

AUTHORS’ REPLY - Lines 194-205: The Materials and Methods Section 2.4 described the recovery and characterization of non-typhoidal Salmonella isolates by following the standardized protocol. Reference #53 (Andrews et al., 2014), cited in the manuscript, documented the PCR primers and the method conditions for the amplification of the invA gene for typing Salmonella.

Reviewer 2 Report

This manuscript describes the characterization of Salmonella isolated by monitoring water in three rivers that empty into the Sea of Cortez. The authors found that Salmonella was more prevalent in the warm seasons, and that it was negatively influenced by pH and salinity.  The authors then used whole genome sequencing to determine serovar, ST, SNP type, and then compare the isolates using SNP trees. Finally the authors identify several conserved nonsynonymous SNPs and then discus a few of them that may be involved in virulence. Overall the study is interesting and the authors could address the following comments:

1. The authors use an SNP tree to separate the isolates into two clades, and state that within a serotype, isolates differed by less than 400 SNPs, while the different clades differed by over 10,000 SNPs. This seems to just restate what we know about serotypes, that for the most part within a serotype, Salmonella are similar, and that between serotypes there are greater differences. Could the authors elaborate a bit more on what they think this is showing about the isolates in the study.

2. The sequence analysis would benefit from further analysis, for example, were any antibiotic resistance genes found in the isolates? Why was this data not presented? Do the authors plan another paper that will further characterize the isolates?

3. The statement at the end of the abstract and in the end of the discussion that this studies identification of "Multiple nonsynonymous SNPs were detected in stiA, sivH and ratA, genes required for Salmonella fitness and survival, and these findings identified relevant markers to potentially develop targeted intervention strategies for this pathogen." is not supported by the data presented. While it is true that conserved SNPs are interesting, I'm not sure how those lead to targeted interventions. I'm not even sure the function of these genes in pathogenicity are known, and even if they are, the SNPs effects would need to be determined to define what is different about these isolates. Even if the function of the gene is known, that does not likely lead to any targeted intervention. While we would like to block the function of virulence factors that does not mean they are "blockable" or that blocking them would reduce pathogenicity, or that this could be a targeted intervention. Please consider explaining how these things would be done, or consider modifying this statement or removing it.

4.  Were the WGS data for the isolates used to search any databases of WGS from human isolates? If this were done it would help determine if these strains are actually causing disease in humans. 

Author Response

REVIEWER 2

  1. REVIEWER’S COMMENT - The authors use an SNP tree to separate the isolates into two clades, and state that within a serotype, isolates differed by less than 400 SNPs, while the different clades differed by over 10,000 SNPs. This seems to just restate what we know about serotypes, that for the most part within a serotype, Salmonella are similar, and that between serotypes there are greater differences. Could the authors elaborate a bit more on what they think this is showing about the isolates in the study.

AUTHORS’ REPLY - Lines 515-520: The present study employed the CSIPhylogeny method for the molecular characterization of the sequenced genomes in the recovered S. enterica isolates. The findings indicated that the recovered isolates were classified in two separate clades. Interestingly, the SNP analysis also revealed a lower intraserovar diversity in the recovered isolates from river water in the Culiacan Valley in Northwestern Mexico when compared to other studies on the genomic classification of S. enterica isolates from aquatic environment other regions in Mexico (Central Mexico). 

To address the request by Reviewer 2 on elaborating on these findings, the statement in lines 514-516 was changed to read “Recently, a report on S. enterica isolates recovered from surface waters in Central Mexico identified a significantly higher intraserovar diversity based on the CSIPhylogeny analysis of highly variable SNPs [23].”

The statement in lines 518-521 was changed to read “the S. enterica isolates from river water in the Culiacan Valley in Northwestern Mexico were closely related with a lower intraserovar diversity and were found to have less than 400 SNP if belonging to the same serovar.”

  1. REVIEWER’S COMMENT - The sequence analysis would benefit from further analysis, for example, were any antibiotic resistance genes found in the isolates? Why was this data not presented? Do the authors plan another paper that will further characterize the isolates?

AUTHORS’ REPLY - The comparative genomic analysis, conducted in the present study, was focused on the identification of SNPs in the core genome for the molecular subtyping of the recovered S. enterica isolates from agricultural river water. Although the initial analysis did not reveal SNPs in antimicrobial resistance genes for distinguishing S. enterica isolates in clade 1 vs clade 2, future studies will further characterize antimicrobial resistance profiles in the recovered S. enterica isolates. 

  1. REVIEWER’S COMMENT - The statement at the end of the abstract and in the end of the discussion that this studies identification of "Multiple nonsynonymous SNPs were detected in stiA, sivH and ratA, genes required for Salmonella fitness and survival, and these findings identified relevant markers to potentially develop targeted intervention strategies for this pathogen." is not supported by the data presented. While it is true that conserved SNPs are interesting, I'm not sure how those lead to targeted interventions. I'm not even sure the function of these genes in pathogenicity are known, and even if they are, the SNPs effects would need to be determined to define what is different about these isolates. Even if the function of the gene is known, that does not likely lead to any targeted intervention. While we would like to block the function of virulence factors that does not mean they are "blockable" or that blocking them would reduce pathogenicity, or that this could be a targeted intervention. Please consider explaining how these things would be done, or consider modifying this statement or removing it.

AUTHORS’ REPLY - Lines 38-40 and 529-533:  The authors gratefully thank Reviewer 2 for the constructive comments on describing the findings in the present study.  To address the request by Reviewer 2, the statement “targeted intervention strategies” was deleted from the Abstract in lines 38-40 and was changed to read “and these findings identified relevant markers to potentially develop improved methods for characterizing this pathogen.”  In lines 529-533, the statements “the effect of these nonsynonymous SNPs” and “to aid in the development of strategies aimed at reducing the morbidity and mortality caused by this enteric human pathogen” were deleted from the manuscript text.  The new statement was changed to read “Future work is aimed at characterizing relevant markers contributing to the fitness and survival of S. enterica in aquatic environments in the Culiacan Valley, an important agricultural region in Northwestern Mexico.”

  1. REVIEWER’S COMMENT - Were the WGS data for the isolates used to search any databases of WGS from human isolates? If this were done it would help determine if these strains are actually causing disease in humans.

AUTHORS’ REPLY - The authors thank Reviewer 2 for the suggested analysis. The objectives of the present study were to employ and optimize an efficient ultrafiltration method for the recovery of Salmonella from agricultural river water and subsequently to evaluate the effect of water physicochemical parameters on the prevalence of Salmonella as well as employ molecular typing methods for the genomic classification of the recovered Salmonella isolates.  Although the genomic comparison of the recovered isolates to human isolates was outside the scope of the present study, future studies will further examine the genome content of the environmental S. enterica isolates to clinical S. enterica isolates from Mexico and other geographical locations.